# A Rapid and Sensitive Gold Nanoparticle-Based Lateral Flow Immunoassay for Chlorantraniliprole in Agricultural and Environmental Samples

**DOI:** 10.3390/foods13020205

**Published:** 2024-01-09

**Authors:** Yanling Wu, Jiao Li, Jie Zhu, Zhaoxian Zhang, Shuguang Zhang, Minghua Wang, Xiude Hua

**Affiliations:** 1College of Plant Protection, Nanjing Agricultural University, Nanjing 210095, China; 2021102123@stu.njau.edu.cn (Y.W.); 2020202060@stu.njau.edu.cn (J.L.); 2022102109@stu.njau.edu.cn (J.Z.); wangmha@njau.edu.cn (M.W.); 2State & Local Joint Engineering Research Center of Green Pesticide Invention and Application, Nanjing 210095, China; 3Key Laboratory of Agri-Food Safety of Anhui Province, College of Resources and Environment, Anhui Agricultural University, Hefei 230036, China; zhangzx@ahau.edu.cn; 4College of Sciences, Nanjing Agricultural University, Nanjing 210095, China; shuguangz@njan.edu.cn

**Keywords:** chlorantraniliprole, monoclonal antibody, lateral flow immunoassay, gold nanoparticle, pesticide residue

## Abstract

Chlorantraniliprole (CAP) is a new type of diamide insecticide that is mainly used to control lepidopteran pests. However, it has been proven to be hazardous to nontarget organisms, and the effects of its residues need to be monitored. In this study, five hybridoma cell lines were developed that produced anti-CAP monoclonal antibodies (mAbs), of which the mAb originating from the cell line 5C5B9 showed the highest sensitivity and was used to develop a gold nanoparticle-based lateral flow immunoassay (AuNP-LFIA) for CAP. The visible limit of detection of the AuNP-LFIA was 1.25 ng/mL, and the detection results were obtained in less than 10 min. The AuNP-LFIA showed no cross-reactivity for CAP analogs, except for tetraniliprole (50%) and cyclaniliprole (5%). In the detection of spiked and blind samples, the accuracy and reliability of the AuNP-LFIA were confirmed by a comparison with spiked concentrations and verified by ultra-performance liquid chromatography–tandem mass spectrometry. Thus, this study provides the core reagents for establishing CAP immunoassays and a AuNP-LFIA for the detection of residual CAP.

## 1. Introduction

Chlorantraniliprole (CAP), 3-bromo-N-[4-chloro-2-methyl-6-[(methylamino)carboxy] phenyl-1-[3-chloro-2-pyridinyl]-1H-pyrazole-5-amide, is an anthranilic diamide insecticide developed to control lepidopterous insects by DuPont Crop Protection in 2008. In just eight years, it has become the top-selling insecticide, with sales of USD 1.3 billion [1]. However, researchers found that beneficial insects could be harmed by eating nectar treated with CAP residue [2]. In addition, CAP has been proven to promote obesity and cardiac problems by affecting crucial obesity genes and disrupting protein secondary structures [3]. Many countries have regulated the residues of CAP in agricultural products with maximum residue limits (MRLs) to ensure food safety and human health. For example, the MRL of CAP ranges from 0.01 to 40 mg/kg for 84 foods in the National Food Safety Standards of China (GB 2763-2021) [4] and 381 foods in the European Union regulation 2022/1343 (EU 2022/1343) [5]. As a result, it is crucial to detect CAP in both food and the surroundings.

Currently, the detection methods for CAP include gas chromatography–mass spectrometry (GC−MS) [6,7,8,9] and liquid chromatography–tandem mass spectrometry (LC−MS/MS) [10,11,12,13,14]. The reported methods are sensitive and exact, but they require costly equipment and time-consuming sample pre-treatment [15].

Immunoassays have been widely used to detect various analytes in biomedicine, food, and the environment because of the advantages of being simple, fast, and economical [16]. Among immunoassays, the enzyme-linked immunosorbent assay (ELISA) is widely considered the most prevalent and mature method [17]. To date, indirect competitive ELISAs (ic-ELISAs) on the basis of anti-CAP monoclonal antibody (mAb) for CAP detection in environmental and agricultural samples have shown half-inhibitory concentrations (IC_50_) of 1.6 ng/mL [18] and 1.5 ng/mL [19]. However, the long reaction time, tedious operating procedures, and the need for specialized equipment have limited the application of ELISAs in field testing [20]. Compared with ELISAs, gold nanoparticle-based lateral flow immunoassays (AuNP-LFIAs) possess the advantages of one-step detection and do not require specialized equipment and direct visual readout [17]. Therefore, they have tremendous potential for on-site rapid testing and have been used to detect various small-molecule pesticides, such as quizalofop-p-ethyl [20], quinclorac [15], and deoxynivalenol [21]. However, no AuNP-LFIA-based monoclonal antibody has been reported for CAP.

In this study, an artificial antigen of CAP was synthesized and used to prepare anti-CAP mAb. The AuNP-LFIA was established for the quick field detection of CAP in environmental samples and agricultural products. The reliability of the AuNP-LFIA was validated by comparing the results of the AuNP-LFIA and ultra-performance liquid chromatography–tandem mass spectrometry (UPLC−MS/MS) for the detection of CAP residues in blind samples of brown rice.

## 2. Materials and Methods

### 2.1. Reagents and Materials

CAP (99.0%), broflanilide (99.3%), cyclaniliprole (96.3%), flubendiamide (98.0%), tetraniliprole (97.5%), and benzamide (98.0%) were bought from the Shanghai Pesticide Research Institute Co., Ltd. (Shanghai, China). 3-Bromo-1-(3-chloropyridin-2-yl)-1H-pyrazole-5-carboxylic acid (99.5%) and 2-amino-3-methyl-5-chlorobenzoic acid (98.0%) were purchased from the Shanghai Haohong Bio-pharmaceutical Science and Technology Co., Ltd. (Shanghai, China). Bovine serum albumin (BSA), ovalbumin (OVA), *N*-hydroxysuccinimide (NHS), *N*,*N*-dimethylformamide (DMF), *N*,*N*’-dicyclohexylcarbodiimide (DCC), chloroauric acid (HAuCl_4_), Freund’s incomplete adjuvant (FIA), and Freund’s complete adjuvant (FCA) were purchased from Sigma Aldrich (Shanghai, China). Tetramethyl benzidine (TMB) and polyethylene glycol-1500 (PEG1500) were purchased from Thermo-Scientific (Rockford, IL, USA). Goat anti-mouse IgG horseradish peroxidase (HPR)-conjugated goat anti-mouse IgG and goat anti-mouse IgG were purchased from Boster (Pleasanton, CA, USA). Nitrocellulose (NC) membranes were purchased from Millipore (Boston, MA, USA). Microplates (96 and 24 wells) for cell culture and ELISAs were purchased from Conning Inc. (Corning, NY, USA). All organic reagents are of analytical purity, and biological reagents were prepared in double distilled water.

### 2.2. Preparation of Hapten and Antigens

Hapten was conjugated by previously reported procedures with some modifications [22]. The process is depicted in Figure 1. In brief, 0.01 mol of 3-bromo-1-(3-chloropyridin-2-yl)-1hydro-pyrazole-5-carboxylic acid was dissolved in 30 mL of acetonitrile, and then 1.22 mL of CH_3_SO_2_Cl and 2.7 mL of pyridine were added dropwise into acetonitrile in turn. After 10 min of incubation at room temperature, 0.015 mol of 2-amino-3-methyl-5-chlorobenzoic acid and 2.7 mL of pyridine were added to the above mixture and stirred for 15 min. A total of 1.22 mL of CH_3_SO_2_Cl was added and stirred at room temperature. The end of the reaction was detected by thin-layer chromatography. After adding 17 g of water, the mixture was filtered and recrystallized using ethanol to obtain the compound 2-[3-bromo-1-(3-chloro-2-pyridinyl)-1H-5-pyrazol]-6-chloro-8-methyl-4H-benzo[d]. A total of 0.26 mmol of aminobutyric acid and 0.45 mmol of NaOH were added to 5 mL of anhydrous DMF in a 25 mL round-bottomed flask and stirred at 25 °C for 1 h. Then, 0.22 mmol of 2-[3-bromo-1-(3-chloro-2-pyridinyl)-1H-5-pyrazol]-6-chloro-8-methyl-4H-benzo[d] [1,3] oxazin-4-one was added. After stirring for 12 h, the solution was mixed with 15 mL of water, adjusted to a pH of 3 using concentrated hydrochloric acid, and then extracted three times with 5 mL of ethyl acetate. Anhydrous sodium sulfate was used to dry the mixture. Recrystallization with hexane and ethyl acetate (3:1, *v*/*v*) produced 80 mg of CAP hapten in 64.7% yield, which was characterized by high-resolution mass spectrometry (HRMS) and nuclear magnetic resonance hydrogen spectroscopy (^1^H NMR).

CAP artificial antigen was synthesized using the active ester method [23]. In brief, 0.1 mmol of CAP hapten ((4-(2-(3-bromo-1-(3-chloro-2-pyridyl)-1H-pyrazole-5-carboxamido)-5-chloro-3-methylbenzoylamino), 0.2 mmol of NHS, and 0.2 mmol of DCC were mixed in 2.5 mL of DMF. Then, after centrifugation for 10 min and stirring in the dark for 12 h, the precipitate was discarded, and the supernatant was added to a phosphate buffer (5 mL) containing 15 mg/mL OVA or BSA and stirred for 4 h at 25 °C. The solution containing the conjugate was finally dialyzed and preserved at −20 °C. The coupling ratio is calculated as follows [24,25]:Coupling ratio = (ε_conjugate_ – ε_protein_)/ε_hapten_(1)

### 2.3. Preparation of anti-CAP mAb

The animal experiments were approved by the Department of Science and Technology of Jiangsu Province (license number SYXK (SU) 2021–0086 and were approved on 14 December 2021). Animal immunization protocols followed the methods described in a previous report [26]. Briefly, hapten-BSA (1 mg/mL) and an equal volume of FCA were emulsified for primary immunization as immunogens and injected into BALB/c mice (6–8 weeks). After three weeks, booster immunizations were given every two weeks with the same amount of FIA-emulsified immunogen used in the injection. After the fifth immunization, the inhibition and titers of antisera were analyzed using ic-ELISA. The selected mouse that had the highest inhibition rate and titer was intraperitoneally injected with immunogen in PBS and killed three days later.

PEG1500 was used to fuse SP2/0 myeloma cells (pre-laboratory storage) with selected spleen cells [27]. After selection by ic-ELISA, positive hybridoma cell lines were obtained by subcloning using the limited dilution method [28]. The ascite antibody of anti-CAP was obtained by intraperitoneal injection of positive hybridoma cells into mice. The collected ascites were incubated at 37 °C for 15 min and then centrifuged, and the supernatant was collected and purified by a protein A column (GE Healthcare, Pittsburgh, PA, USA) to obtain the monoclonal antibody. After dialysis with saline, the sample was verified by SDS-PAGE and preserved at −20 °C.

### 2.4. Evaluation of Anti-CAP mAb

The sensitivity of the anti-CAP mAb was evaluated using ic-ELISA. The antigen (hapten-OVA, 250 μg/mL) in a bicarbonate buffer was poured into a microtiter plate at 100 μL/well and incubated at 37 °C for 2 h. PBS containing 0.05% Tween-20 (PBST) was used to wash the plate 5 times, and then 3% skim milk in PBS (200 μL/well) was added for blocking. After 1.5 h of incubation at 37 °C, each well was washed 5 times. CAP standard solution (50 μL/well) and an equal volume of mAb (3.9 μg/mL) in PBS were added for 1 h of incubation at 37 °C and then washed 5 times with PBST. Then, the HRP-labeled goat anti-mouse IgG antibody (1:20,000) was diluted in PBS, 100 μL/well was added, and the cells were incubated for 1 h at 37 °C. After washing 5 times with PBST, 100 μL/well fresh substrate solution (100 μL of 10 mg/mL TMB and 32 μL of 0.75% H_2_O_2_ per 10 mL of acetate buffer) was added and incubated at 37 °C for 15 min. To terminate the reaction, 50 μL of 2 mol/L H_2_SO_4_ (50 μL/well) was added per well. A Spectra-Max M5 reader (Molecular Devices, San Francisco, CA, USA) was used to measure the absorbance at 450 nm. Origin software (version, 10.0.5.157 ) was used to generate a standard curve and calculate the IC_50_, and the antibody affinity constant K_aff_ was determined by an ELISA [29].

The specificity of the mAb was assessed by detecting serial concentrations of CAP analogs using ic-ELISA. The cross-reactivity (CR) was calculated by the formula as follows.
CR% = (IC_50_ values of CAP/IC_50_ values of analog) × 100%(2)

### 2.5. Preparation of the AuNP-Labeled mAb Probe

Gold nanoparticle-labeled mAb probes are critical biosensors for the AuNP-LFIA and were prepared by coupling antibodies against CAP to gold nanoparticles (AuNPs). A reduction in HAuCl_4_ by trisodium citrate was used to prepare AuNPs [30]. The pH of the AuNPs was 8.2 by 0.1 mol/L K_2_CO_3,_ and 37.5 μg of mAb per 1 mL AuNPs was added. The reaction solution was stirred at room temperature. After stirring for 30 min, 10% sodium borate solution was added to obtain a final concentration of 1%. After the solution was stirred at room temperature for 30 min, it was centrifuged at 8000 rpm and 4 °C for 15 min, and the supernatant was discarded. The red precipitate was redissolved in 1/10 of the original volume of 0.01 mol/L sodium borate buffer solution containing 3% sucrose and 2% BSA and preserved at 4 °C.

The coupling ratio of mAb on the surface of AuNPs was calculated as follows: coupling ratio = (m1 × *N_A_*/M1)/[m2/(ρAuNP × V)], where m1 and m2 represent the masses of the mAb in AuNP-labeled mAb (measured by SDS-PAGE) and the AuNP, respectively, M1 represents the molecular weight of the antibody as 150 kDa, *N_A_* represents Avogadro’s constant 6.02 × 10^23^, V is the volume of single AuNP, and ρAuNP = 1.97 × 10^7^ g/m^3^ [31].

### 2.6. Preparation of AuNP-LFIA Strips

As illustrated in Figure 2a, the AuNP-LFIA strip consists of five elements: the sample pad, conjugate pad, absorbent pad, polyethylene floor, and NC membrane. The NC membrane was sprayed with hapten-OVA at a concentration of 1.25 mg/mL, which was labeled as the test line (T line), and goat anti-mouse IgG antibody at 0.125 mg/mL, designated as the control line (C line). After drying for 1 h at 37 °C, a conjugate pad was immobilized with the AuNP-labeled mAb (5 μL/cm) and dried at 37 °C for 1 h. The NC membrane, absorbent pad, conjugate pad, and sample pad were then affixed to a polyethylene floor, cut into 4.0 mm wide AuNP-LFIA strips, and preserved in a dry place and at room temperature.

### 2.7. Procedure of the AuNP-LFIA

CAP standard solution or sample solution (100 μL) was dropped into the prepared AuNP-LFIA sample pad, and the AuNP-LFIA provided results within 10 min through capillary action and immune reaction. After incubation for 10 min, the results can be evaluated visually. As Figure 2a shows, the results of the AuNP-LFIA can be divided into three cases. (1) When there is no color on the C line, the results are considered invalid whether or not there is color on the T line. (2) When there is no CAP in the sample or its concentration is below the limit of detection (LOD), the AuNP-labeled mAb would bind to the coating antigen on the T line and the goat anti-mouse IgG on the C line, which generates the darker red T line and the red C line. The result is interpreted as negative (−). (3) With an increase in CAP concentration, the binding between AuNP-labeled mAb and coating antigen is gradually inhibited by CAP, so the red T line would gradually become lighter. When the T line becomes significantly lighter than the C line or disappears, the result could be interpreted as positive (+).

### 2.8. Detection of Spiked Samples

Brown rice, apples, and cabbage were acquired from a supermarket in Nanjing and stored at −20 °C after chopping and homogenizing. The soil was collected locally from Nanjing. Samples (10 g) were added to 50 mL centrifuge tubes. CAP standard buffer solutions were added to soil, brown rice, and apple samples at terminal concentrations of 0, 0.025, 0.05, 0.1, and 0.2 mg/kg, and CAP standard buffer solutions were added to Chinese cabbage samples at terminal concentrations of 0, 0.05, 0.1, 0.2, and 0.4 mg/kg. After standing for 2 h, 20 mL of 60% methanol-PBS was added, and the mixture was vortexed, sonicated, vortexed again for 5 min, and then centrifuged for 5 min at 4000 rpm. The supernatant was assayed after appropriate dilution. Each sample was spiked and analyzed in triplicate.

### 2.9. Verification by UPLC−MS/MS

Ten blind samples of brown rice containing CAP were simultaneously analyzed by UPLC−MS/MS and the AuNP-LFIA. For the AuNP-LFIA, the blind samples were determined according to the spiked samples. For UPLC−MS/MS, a 10 g homogenized sample was extracted by 20 mL of formic acid water/acetonitrile (1:1, *v*/*v*) with shaking for 15 min. Then, 2.5 g NaCl was added, vortexed for 5 min, and centrifuged at 4000 rpm for 5 min. The supernatant (3 mL) was mixed with 75 mg of C18 and 150 mg of N-propylethylenediamine and vortexed for 5 min. After centrifugation at 4000 rpm for 5 min, the supernatant was passed through a 0.22 μm filter. Finally, 5 μL supernatant was analyzed by UPLC−MS/MS (Waters, Milford, CT, USA). The analysis was based on a BEH C18 column (2.1 mm × 50 mm, 1.7 μm) (Agilent, Wilmington, NC, USA). The mobile phase mixture was 0.1% formic acid water/acetonitrile (4/6, *v*/*v*). The flow rate of the column at 25 °C was 0.2 mL/min. Under the above conditions, the retention time was 1.4 min for CAP. Three replicates per sample were performed.

## 3. Results and Discussion

### 3.1. Characterization of Hapten and Antigens

CAP hapten was characterized by ESI-MS and ^1^H NMR. The *m*/*z* value of the hapten was 553. ^1^H NMR (DMSO) results: 1.62 (m, CH_2_, 2H), 2.11 (s, CH_3_, 3H), 2.46 (t, CH_2_, 2H), 3.10 (m, CH_2_, 2H), 7.34–8.30 (m, aromatic-H, 6H), 8.45 (t, CONHCH_2_, 1H), 10.21 (s, CONH, 1H), and 12.01 (s, COOH, 1H) (Appendix A).

As shown in Appendix A, there is an absorption peak shift in the UV absorption spectrum of the antigen compared to the carrier protein, which indicates that the hapten has been coupled successfully to the carrier protein. The coupling ratios of hapten-BSA and hapten-OVA were 15.1:1 and 10:1, respectively.

### 3.2. Characterization of anti-CAP mAb

Five hybridoma cell lines were obtained that stably secreted anti-CAP mAb, designated 3G1D10, 2C12D11, 5C5B9, 3H12C8, and 1E11E7. Among these, the mAb secreted by the 5C5B9 cell line showed the most sensitivity (Appendix A). The purified mAb 5C5B9 was verified by SDS-PAGE gel analysis (Appendix A), which showed the typical light chain band (~25 kDa) and heavy chain band (~50 kDa), and no other unexpected band. The affinity constant K_aff_ of 5C5B9 determined by the ELISA was 2.8 × 10^10^ L/mol (Appendix A). The standard curve is shown in Appendix A, and the IC_50_ value, detection range (IC_10_–IC_90_), and LOD (IC_10_) were 0.24 ng/mL, 0.008–9.58 ng/mL, and 0.008 ng/mL, respectively. Compared with the reported anti-CAP mAb (IC_50_, 1.6 ng/mL [18] and 1.5 ng/mL [19]), the sensitivity was improved 6.7-fold and 6.3-fold. Only cyclaniliprole and tetraniliprole showed CRs of 28.8% and 0.3%, respectively, because they have an antraniloiyl group (Appendix A), and were similar to those reported previously (CR for cyclaniliprole was 37.5% [19,32].

### 3.3. Sensitivity of the AuNP-LFIA

As shown in Appendix A, SDS-PAGE shows a 50 kDa heavy chain and a 25 kDa light chain of the antibody. The grey value of the bands showed a good positive correlation with the quality of the antibody (Appendix A). The mass of antibody on the surface of 0.54 mg AuNPs was 3.65 μg. Thus, the coupling ratio of AuNP-labeled mAb was 2.5:1. To enhance the sensitivity of the AuNP-LFIA for CAP detection, the parameters of pH, Na^+^, Tween-20 content, organic solvent, and detection time were optimized. As shown in Appendix A, when the Tween-20 concentration was above 0.05%, a positive result was obtained when analyzing the CAP-free solution (false positive). Moreover, the highest sensitivity was found at a pH of 7.4 and 0.07 mol/L Na^+^. Methanol had the least impact on the AuNP-LFIA sensitivity among the organic reagents. However, the LOD of the AuNP-LFIA increases while the sensitivity decreases when the methanol content is higher than 5%. So, the optimal buffer for the AuNP-LFIA was pH 7.4, 0.07 mol/L PBS containing 0.05% Tween-20, and 5% methanol. Additionally, there was no change in the color intensity of the T line and C line when the detection time exceeded 10 min, indicating that the reaction was completed at 10 min. So, the optimal detection time was determined to be 10 min.

Under the optimal conditions, the AuNP-LFIA detection results for a range of CAP concentrations are shown in Figure 3a. Positive test results were obtained when CAP concentrations were above or equal to 1.25 ng/mL. In comparison, they were negative when CAP concentrations were below 1.25 ng/mL. Therefore, the visible LOD of the AuNP-LFIA for CAP detection was 1.25 ng/mL. In addition, the results of the AuNP-LFIA were processed by Image J software (version, 1.8.0) to measure the gray value of the T line. The standard curve of the AuNP-LFIA in quantitative detection is shown in Figure 3b; the linear equation was y = −6.2427x + 0.7463 (R^2^ = 0.9932) and the IC_50_, detection range (IC_10_–IC_90_) and LOD (IC_10_) were 1.23 ng/mL, 0.53–2.85 ng/mL, and 0.53 ng/mL, respectively.

The comparison between the AuNP-LFIA developed in this study and other methods for CAP is shown in Appendix A. The sensitivity of the AuNP-LFIA in this study (visible LOD was 1.25 ng/mL) was improved 2-fold compared to the LFIA reported previously (visible LOD was 2.5 ng/mL) [32]. Although the proposed AuNP-LFIA had similar sensitivity to ELISAs [18,19], the detection time was reduced by more than 13-fold. In addition, compared to LC–MS/MS [33], the AuNP-LFIA did not require specialized equipment, professional personnel, or a complex pre-treatment process, making it more suitable for on-site detection of CAP.

### 3.4. Specificity of the AuNP-LFIA

As shown in Figure 3c, the standard solution of CAP and its structural analogs (1000 ng/mL) were analyzed by AuNP-LIFA. The results after quantification are shown in Figure 3d. Only CAP, tetraniliprole, and cyclaniliprole results were positive (+). As shown in Appendix A, the visible LODs of tetraniliprole and cyclaniliprole are 25 ng/mL and 2.5 ng/mL, respectively. The formula for calculating the CR of the AuNP-LFIA was as follows: (visible LOD of CAP)/(visible LOD of analog) × 100%. Therefore, the CRs were 5% for tetraniliprole and 50% for cyclaniliprole using the AuNP-LFIA, which were similar to the CR determined by ic-ELISA.

### 3.5. Robustness of the AuNP-LFIA

According to the International Union of Pure and Applied Chemistry Standard (IUPAC) [34], intraday experiments were carried out by two operators to test the same standard solution of CAP on the same day, while interday experiments were performed by the same operator to test the same standard solution of CAP at the same time on two consecutive days. Three replicates were performed for each concentration and assayed three times. As shown in Appendix A, the results were consistent at the intraday test and interday test. In addition, the *t*-test results (0.325 for intraday and 0.822 for interday) were greater than 0.05 (*p* > 0.05), so there was no significant difference between the intraday test and interday test. These results indicate that the AuNP-LFIA has good robustness.

### 3.6. Analysis of Spiked Samples

The matrices of samples can affect the binding of antibodies and antigens, which affects the accuracy, stability, and sensitivity of immunoassays. To minimize matrix interference, the dilution of extracts with a buffer is the most commonly used method to eliminate matrix effects. As shown in Figure 4, when the extracts of brown rice, apple, soil, and Chinese cabbage were diluted equal to or more than 12-fold, 12-fold, 12-fold, and 24-fold (the total dilution was 24-fold, 24-fold, 24-fold, and 48-fold, containing 2-fold during extraction), respectively, the detection results for the CAP standard during the matrix dilution were comparable to those acquired in the matrix-free buffer.

After dilution, the result of visual detection in Table 1 showed that positive results were obtained when the spiked concentration in Chinese cabbage samples was higher than or equal to 0.1 mg/kg and when the spiked concentrations in brown rice, apples, and soil were greater than or equal to 0.05 mg/mL. Therefore, the visible LODs of the AuNP-LFIA for CAP in brown rice, apples, soil, and Chinese cabbage were 0.05, 0.05, 0.05, and 0.1 mg/kg, respectively. According to Commission Regulation (EU) 2022/1343 [5], MRLs of CAP in brown rice, apples, and Chinese cabbage were 0.4, 0.4, and 20 mg/kg, respectively. In the National Food Safety Standard of China (GB 2763-2021), the MRLs for CAP in brown rice, apples, and Chinese cabbage were 0.5, 2, and 20 mg/kg, respectively. The visible LODs of the AuNP-LFIA for CAP in brown rice, apples, and Chinese cabbage were all lower than the MRLs in the EU and China, which indicated that the AuNP-LFIA meets the requirements for the detection of CAP in agricultural products. According to previous reports [35], the quantitative results in Table 1 showed that the average recoveries of soil, brown rice, and apple samples at spiked concentrations of 0.025–0.2 mg/kg were in the range of 65.8–71.2% with relative standard deviations (RSDs) of 1.9–6.7% and 61.2–70.7%, 2.5–6.5% and 64.0–71.1%, and 2.9–4.6%, respectively. The average recoveries of cabbage samples were 59.7–65.8% with RSDs of 3.2–6.8% at the spiked concentrations of 0.05–0.4 mg/kg. The average recoveries of the spiked samples did not meet the standards of IUPAC (IUPAC, 70–120% with RSD ≤ 20%) [34], and the results of the *t*-test (|t| > |t|Critical,_2_ = 4.30) showed that there are significant differences between the measured concentrations and spiked concentrations. Therefore, the proposed AuNP-LFIA is more suitable for qualitative detection. However, it is worth noting that the AuNP-LFIA with the advantages of rapid and simple operation is an ideal choice for on-site detection.

### 3.7. Validation by UPLC−MS/MS

The reliability of the AuNP-LFIA was verified by the consistency of the results with UPLC–MS/MS. Serial concentrations of standard solutions prepared by methanol and the extract of blank samples were analyzed by UPLC−MS/MS. The results showed that there was a good linear relationship between the concentration and peak area of CAP in the range of 0.001 mg/kg to 0.5 mg/kg (Appendix A). The slope ratio of the matrix line to the solvent line was 0.51 (lower than 0.8), indicating a signal suppression effect in the matrix effect. So, the concentration of CAP in brown rice samples was calculated by the matrix standard curve. The results showed that the average recoveries of CAP at the spiked concentrations of 0.005–0.5 mg/kg ranged from 94.6% to 99.9%, with RSDs of 2.7–6.1% (Appendix A). These results meet the requirements of US Environmental Protection Agency standards (EPA, 70–120% with RSD ≤ 20%) [36], which indicates that the UPLC−MS/MS method can quantitatively and accurately detect CAP in brown rice samples. The limit of quantification (LOQ) for CAP in brown rice samples by UPLC−MS/MS was the lowest spiked concentration of 0.005 mg/kg. Ten samples of brown rice spiked with unknown concentrations of CAP were simultaneously detected using the AuNP-LFIA and UPLC−MS/MS. As shown in Table 2, four negative samples (sample Nos. 1 to 4) and six positive samples (sample Nos. 5 to 10) were determined by the AuNP-LFIA. The concentrations of CAP in the negative samples (sample Nos. 1 to 4) detected by UPLC−MS/MS were lower than the visible LOD in brown rice (0.05 mg/kg), while the concentrations in the positive samples (sample Nos. 5 to 10) were higher than the visible LOD. Therefore, the detection results of the AuNP-LFIA were in agreement with UPLC−MS/MS, which validates the reliability of the AuNP-LFIA.

## 4. Conclusions

In summary, an anti-CAP mAb was produced with high sensitivity and specificity, and a AuNP-LFIA was prepared to analyze CAP in environmental and agricultural products. The results can be visually determined within 10 min, and the visible LOD of the AuNP-LFIA was 1.25 ng/mL. The visible LODs of the AuNP-LFIA for CAP in brown rice, apples, soil, and Chinese cabbage were 0.05, 0.05, 0.05, and 0.1 mg/kg, respectively, which were lower than the MRLs of CAP in the agricultural products. Therefore, the sensitivity of the AuNP-LFIA fits the requirements for the detection of CAP in the samples. In addition, the AuNP-LFIA results for the brown rice samples spiked with unknown concentrations of CAP were concordant with UPLC−MS/MS, which demonstrates that the AuNP-LFIA is accurate and reliable. Overall, this study provided a high-sensitivity anti-CAP mAb as the core reagent for the establishment of CAP immunoassays and a AuNP-LFIA method for the rapid detection of CAP in environmental and agricultural samples.

## Figures and Tables

**Figure 1 foods-13-00205-f001:**
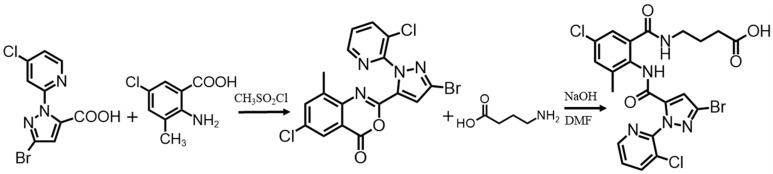
Synthesis routes of CAP hapten.

**Figure 2 foods-13-00205-f002:**
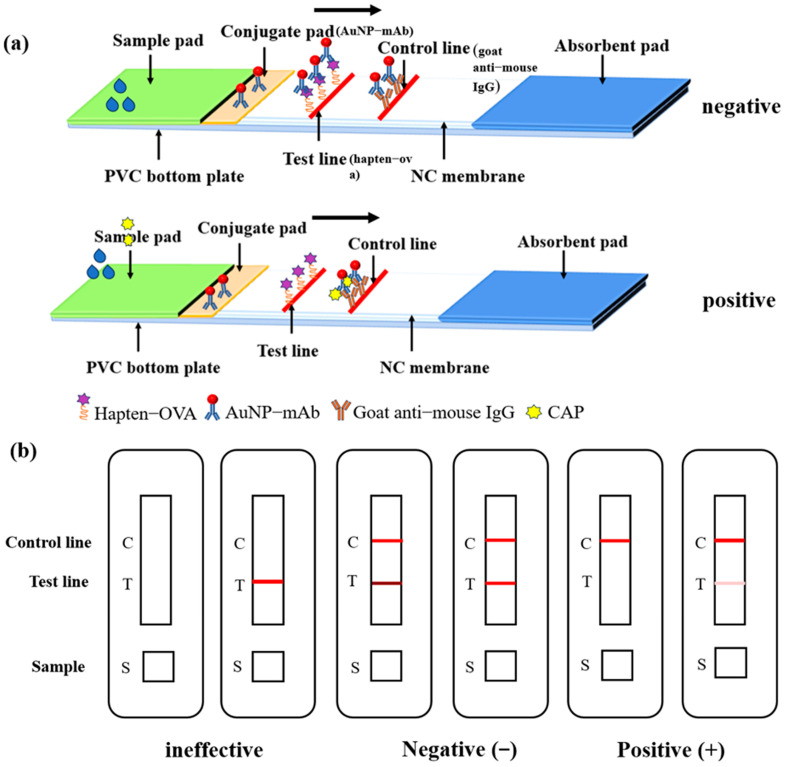
Schematic diagrams of the AuNP-LFIA. (**a**) The binding between the AuNP-labeled mAb and hapten-OVA would generate a red color on the T line for the negative sample, which would be inhibited by CAP in positive samples. (**b**) Result judgment of the AuNP-LFIA.

**Figure 3 foods-13-00205-f003:**
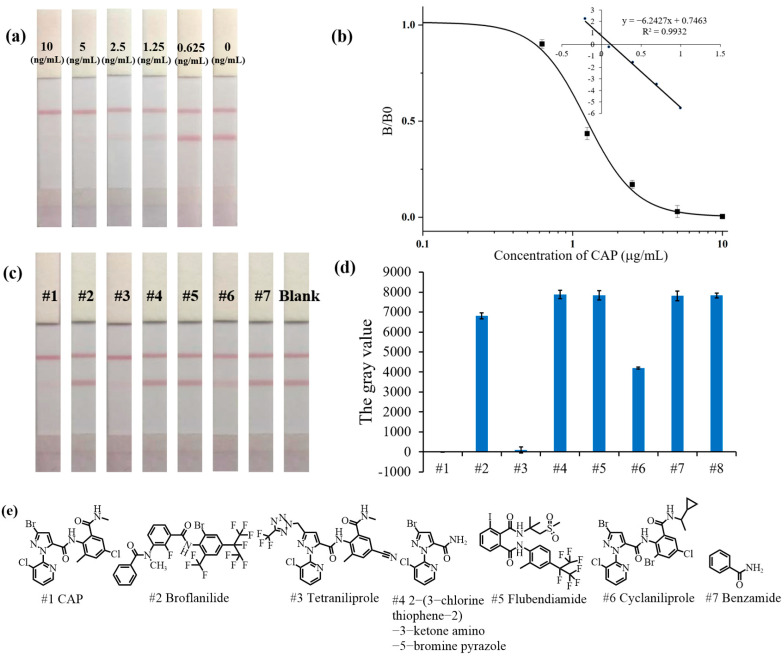
(**a**) the sensitivity of the AuNP-LFIA for CAP. (**b**) Standard curve of the intensity of the T line versus CAP concentration. (**c**,**d**) Visual detection results and quantification of CAP and its structural analogs (1000 ng/mL) by the AuNP-LFIA. (**e**) Structural formulas of CAP and its analogs.

**Figure 4 foods-13-00205-f004:**
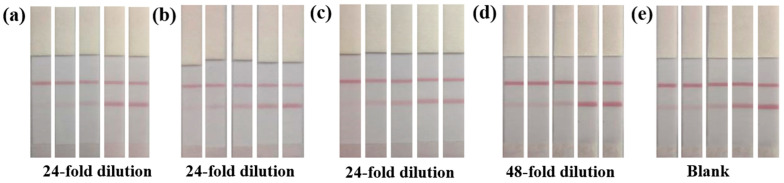
Images of the AuNP-LFIA for CAP detection in different sample matrices with dilution, including brown rice (**a**), apple (**b**), soil (**c**), Chinese cabbage (**d**), and a blank (working buffer) (**e**). (CAP concentrations from left to right were 5, 2.5, 1.25, 0.625, and 0 ng/mL, respectively).

**Table 1 foods-13-00205-t001:** Results of CAP residue detection by the AuNP-LFIA on spiked samples (*n* = 3).

Sample	Spiked (mg/kg)	Visualization Results			Quantitative Results (mg/kg)	Recovery (%)	RSD (%)
Soil	0	− − − ^a^	− − −	− − −	<LOD	/ ^c^	/
	0.025	− − −	− − −	− − −	0.016 ± 0.001	65.9	6.7
	0.05	+ + + ^b^	+ + +	+ + +	0.034 ± 0.002	68.0	5.9
	0.1	+ + +	+ + +	+ + +	0.071 ± 0.002	71.2	2.3
	0.2	+ + +	+ + +	+ + +	0.132 ± 0.003	65.8	1.9
Brown rice	0	− − −	− − −	− − −	<LOD	/	/
	0.025	− − −	− − −	− − −	0.015 ± 0.001	61.2	6.5
	0.05	+ + +	+ + +	+ + +	0.033 ± 0.002	65.5	4.9
	0.1	+ + +	+ + +	+ + +	0.065 ± 0.002	65.3	2.5
	0.2	+ + +	+ + +	+ + +	0.141 ± 0.006	70.7	4.0
Apple	0	− − −	− − −	− − −	<LOD	/	/
	0.025	− − −	− − −	− − −	0.016 ± 0.001	64.0	3.2
	0.05	+ + +	+ + +	+ + +	0.035 ± 0.001	70.0	4.6
	0.1	+ + +	+ + +	+ + +	0.071 ± 0.002	71.1	2.9
	0.2	+ + +	+ + +	+ + +	0.133 ± 0.004	66.3	3.1
Chinese cabbage	0	− − −	− − −	− − −	<LOD	/	/
	0.05	− − −	− − −	− − −	0.031 ± 0.002	61.5	6.1
	0.1	+ + +	+ + +	+ + +	0.065 ± 0.002	65.6	3.2
	0.2	+ + +	+ + +	+ + +	0.131 ± 0.005	65.8	3.9
	0.4	+ + +	+ + +	+ + +	0.238 ± 0.016	59.7	6.8

^a^ “−” represents negative results, ^b^ “+” represents positive results, and ^c^ “/” represents not calculated.

**Table 2 foods-13-00205-t002:** The detected results using the AuNP-LFIA and UPLC–MA/MS for brown rice samples (*n* = 3).

Sample No.	UPLC–MS/MS (mg/kg)	AuNP-LFIA ^a^	Sample No.	UPLC–MS/MS (mg/kg)	AuNP-LFIA ^a^
1	<LOQ		6	0.062	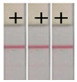
2	0.005	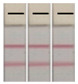	7	0.071	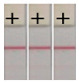
3	0.009	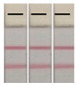	8	0.14	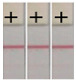
4	0.018	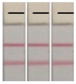	9	0.28	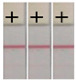
5	0.053	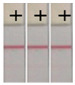	10	0.48	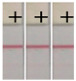

^a^ “+” represents positive detection results and “−” represents negative detection results.

## Data Availability

Data are contained within the article.

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
