# Peer review of "A Rapid and Sensitive Gold Nanoparticle-Based Lateral Flow Immunoassay for Chlorantraniliprole in Agricultural and Environmental Samples"

_foods, 2024, doi:10.3390/foods13020205_

Round 1
Reviewer 1 Report
Comments and Suggestions for Authors
The authors must demonstrate that in the cited extraction conditions chlorantraniliprole is extracted quantitatively. Otherwise, the sample analysis is useless.
The calculations about how LOD and LOQ were calculated should be explained.
In the figure caption of Fig S2, concentrations should be included.
Figure S4 is useless
Comments on the Quality of English Language
The title is not attractive to the readers at all.
English should be revised. i.e. Line 178: “Repeated three times per sample.”
Reviewer 2 Report
Comments and Suggestions for Authors
In this study, five hybridoma cell lines were developed 13 that produced anti-CAP monoclonal antibodies (mAbs), of which the mAb originating from the cell 14 Line 5C5B9 showed the highest sensitivity and were used to develop a gold nanoparticle-based lateral flow immunoassay (AuNP-LFIA) for CAP. The results aren't confirmed with sufficient information, and not all of it is well-described. There are several points which the author may like to consider in order further to improve the quality of the manuscript that has to be discussed as follows. Please resolve these issues.
1. The authors need to calculate the affinity value of the Ab with the target.
2. Authors should calculate the amount of Ab on the surface AuNPs.
3. All analytical performance provided in the figure need to be discussed more thoroughly. For instance, please discuss why the %recovery obtained is good enough (probably with standard t-test compare the spiked concentrations to the detectable concentrations) and why obtained %RSD provided good results. For assess this point please see the following reference and cited. Analytica Chimica Acta 2023, 1252, 341073
4. Please investigate the interday and intraday measurements and report with standard t-test results with confident level of 95% (alpha = 0.05), ensuring the robustness of the developed method
Reviewer 3 Report
Comments and Suggestions for Authors
The presented work is devoted to preparation of monoclonal antibody to chlorantraniliprole and fabrication of LFIA strip test for the determination of this insecticide in several matrixes, namely rice, apple, soil, and Chinese cabbage. The work is appropriately planned and clearly written, but lacks of novelty and originality, qualitative improvements and new matrixes. It mainly duplicates and reproduces procedures and methods described earlier [16, 22, 29]. In my opinion, the presented article does not correspond to the level of high-ranking FOODS publications.
Additional minor points.
1. Fig 1. Error in 2-[3-bromo-1-(3-chloro-2-pyridinyl)-1H-5-pyrazol]-6-chloro-8-methyl-4H-benzo[d] [1,3] oxazin-4-one formula
2. The issue of this substance is not indicated
3. L 208. The difference between 0.24 ng/mL and 1.6 ng/mL [16] is 6.7 times instead of 8-fold.
4. L 258 Typo “flod” instead of fold
Comments on the Quality of English Language
Minor editing of English language required
Reviewer 4 Report
Comments and Suggestions for Authors
- In Figure 2a (scheme), the authors should illustrate what they put in the conjugate pad, what is on the test and control lines, and how the color signal appears with positive and negative samples. Furthermore, the outcome in figure 2b does not correspond to your technique concept. What is the difference between the first negative result image and the second positive result image?
- Should the authors provide a figure confirming the synthesis and purification of mAb?
- This LFA's reaction time should be optimized to highlight target detection time.
- I suggest that the authors use tools like ImageJ to determine color intensity and convert qualification to quantification. The quantitative readout of these image results should be included and the detection limit calculated using the standard curve for detecting CAP
- Table 1 should be transformed into a quantitative result with Recovery% and RSD% -
Round 2
Reviewer 2 Report
Comments and Suggestions for Authors
It seems that the authors have answered all the questions completely and comprehensively and the article has become much more fruitful from the scientific point of view.
Author Response
Thank you very much for your comments and suggestions regarding our manuscript.
Reviewer 3 Report
Comments and Suggestions for Authors
After the revision, the article began to look better, the authors responded to comments and corrected some shortcomings. However, the article has not acquired any novelty (compared with https://doi.org/10.1039/D2AN01366E)and does not demonstrate significant improvement (neither sensitivity nor selectivity). This, in my opinion, does not meet the high requirements of Foods
The scales on the tab in Fig. 3b is not indicated.
Reviewer 4 Report
Comments and Suggestions for Authors
I appreciate your efforts to enhance the manuscript's content.
Author Response

(The authors gave the same response as above.)
